# Pilot randomised controlled trial of the Risk Acceptance Ladder (RAL) as a tool for targeting health communications

**Olga Perski**, **Claire Stevens**, **Robert West**, **Lion Shahab***

Research Department of Behavioural Science and Health, University College London, London, United Kingdom

* lion.shahab@ucl.ac.uk

## Abstract

### Background

Improving adherence to self-protective behaviours is a public health priority. We aimed to assess the potential effectiveness and ease of use of an online version of the Risk Acceptance Ladder (RAL) in promoting help-seeking for cigarette smoking, excessive alcohol consumption, insufficient physical activity, or low fruit and vegetable consumption.

### Methods

843 UK adults were recruited, of whom 602 engaged in at least one risky behaviour. Those with no immediate plans to change ($n$ = 171) completed a behaviour specific RAL. Participants were randomised to one of two conditions; a short message congruent (on-target, $n$ = 73) or incongruent (off-target, $n$ = 98) with their RAL response. Performance of the RAL was assessed by participants' ability to select an applicable RAL item and reported ease of use of the RAL. Effectiveness was assessed by whether or not participants clicked a link to receive information about changing their target behaviour.

### Results

Two thirds (68.9%, 95% CI = 61.8%-75.3%) of participants were able to select an applicable RAL item that corresponded to what they believed would need to change in order to alter their target behaviour, with 64.9% (95% CI = 57.5%-71.7%) reporting that it was easy to select one option. Compared with the off-target group, participants allocated to the on-target group had greater odds of clicking on the link to receive information (31.5% vs 19.4%; OR = 2.07, 95% CI = 1.01–4.26).

### Conclusion

The Risk Acceptance Ladder may have utility as a tool for tailoring messages to prompt initial steps to engaging in self-protective behaviours.

**Data Availability Statement:** The quantitative dataset underpinning the analyses is available here: https://doi.org/10.6084/m9.figshare.12942053.v1. Qualitative, open-ended responses are available

upon request from the corresponding author (lion.shahab@ucl.ac.uk).

**Funding:** OP receives salary support from Cancer Research UK (C1417/A22962). OP and LS are members of SPECTRUM, a UK Prevention Research Partnership Consortium (MR/S037519/1). UKPRP is an initiative funded by the UK Research and Innovation Councils, the Department of Health and Social Care (England) and the UK devolved administrations, and leading health research charities.

**Competing interests:** OP and CS report no conflicts of interest. RW has undertaken research and consultancy for and receives travel funds and hospitality from manufacturers of smoking cessation medications (Pfizer, GlaxoSmithKline and Johnson and Johnson). LS has received a research grant and honoraria for a talk and travel expenses from manufacturers of smoking cessation medications (Pfizer and Johnson & Johnson). This does not alter our adherence to PLOS ONE policies on sharing data and materials.

## Introduction

Improving adherence to self-protective behaviours, or reducing harmful behaviours, is an important goal for public health [1]. This includes, but is not limited to, stopping tobacco use, reducing alcohol consumption, improving diet and increasing physical activity [2–4]. A commonly used framework for the systematic development of behaviour change interventions is the Behaviour Change Wheel [5]. A key process in this framework is identifying which aspects of someone's capability, opportunity and/or motivation need to change in order for the behaviour to change [6]. This paper describes a preliminary evaluation of a self-report measure, the Risk Acceptance Ladder (RAL) [7–9], that aims to establish what aspects of capability, opportunity and/or motivation to focus on in a behaviour change intervention to prompt someone to take an initial step in making the change.

There has been a large amount of research on tailoring behaviour change interventions to individual characteristics. A commonly used model, the transtheoretical model, has been used to tailor interventions according to a putative stage in the change process: 'precontemplation', 'contemplation', 'preparation', 'action', and 'maintenance' [10, 11]. There is mixed evidence that stage-matched interventions are more effective than mismatched ones [12, 13] and there is also evidence that interventions that ignore stage matching can be more effective than ones that seek to identify the stage of change and only offer support to people who show an interest [14–16].

The Risk Acceptance Ladder (RAL) was developed with the idea that people might themselves have some level of insight into what would be required for them to change their behaviour [7–9]. Using the Capability-Opportunity-Motivation-Behaviour (COM-B) model as basis, the RAL proposes that there might be a natural hierarchy of factors leading to the current risky behaviour. The person may never have heard that it was risky, may have heard about it but not understood the message, may have understood it but not believed it, may have believed it but not been concerned about it, may have been concerned but not enough to outweigh the perceived benefits of the risky behaviour, or may have been sufficiently concerned but found it difficult for a number of external or internal reasons. If it turns out that people have some insight into what is preventing change, and this can be classified hierarchically, a brief questionnaire may provide a useful starting point for targeting interventions to initiate change. This is an unknown and so it was important to undertake a preliminary evaluation of the RAL.

This study aimed to evaluate an online version of the RAL focusing on four important health-related behaviours: smoking, alcohol consumption, diet and physical activity. In principle, the approach could be used for other health behaviours such as risky driving, infection control or sexual health behaviours. The choice of an online test of the tool was motivated by the fact that, if it was shown to have some value, it would be easy to implement through websites and online platforms, and also that it was possible to establish an easily measurable behavioural response in terms of 'clicking through' to a page that would represent a first stage in the change process.

The research questions addressed by the current study were:

1. To assess performance: How readily can respondents choose a single 'rung' of the Risk Acceptance Ladder as a possible target for change?

2. To assess effectiveness: Does messaging that directly addresses the selected 'rung' (on-target messaging) lead to a greater likelihood of taking an initial step in making the change than messaging that addresses a different 'rung' (off-target messaging)?

## Methods

### Study design and setting

This was a pilot, parallel group, randomised controlled trial (RCT) conducted in the UK. The Paper Authoring Tool (https://www.addictionpat.org/) was used in the writing of this report.

### Inclusion criteria

A tiered eligibility procedure was employed (see Fig 1). To be eligible to take part, participants had to reside in the UK, be aged 18+ years, engage in at least one unhealthy behaviour (i.e. cigarette smoking, daily or almost daily alcohol consumption, lack of daily physical activity of at least 30 minutes, or eating less than five portions of fruit or vegetables daily) and have no immediate plans to change their behaviour.

### Sample recruitment

A link to the study was sent to students at a large UK university via a monthly e-newsletter and the study was advertised on four websites which allow researchers to connect with potential participants. In addition, a pay-per-click advert was posted on Facebook and a link to the study website was shared by members of the research team on social media platforms, including Twitter. Due to the recruitment strategies used, it was not possible to determine how many people were reached by the recruitment methods and to estimate a response rate.

### Procedure

A website was built that enabled participants to take part in the research using either a computer or a mobile device. First, information about the research was provided and participants were asked to provide informed consent. Consenting participants were presented with an online survey to determine eligibility. Those who indicated that they engaged in at least one unhealthy behaviour were assigned a target for change. Participants who engaged in more than one unhealthy behaviour were randomly assigned to a single target behaviour by a computer algorithm. Next, participants were asked about plans to change their behaviour. Those with no immediate plans to change were presented the RAL (see Table 1) and asked to select one statement that most closely described what they believed would need to change in order to change their target behaviour. Participants who completed the RAL were subsequently individually randomised using computer-generated random numbers on a 50–50 basis to one of the two intervention conditions.

Following intervention delivery, participants were thanked for taking part and were shown a web-link that they could visit for more information about changing their target behaviour. Participants who provided contact details were entered into a prize draw to win one of four £50 vouchers. Data were collected between May and December 2015. The study was approved by the university's research ethics committee (Project ID: 6692/001).

### Intervention

Participants allocated to the on-target messaging condition received a targeted message which reflected their individual response to the RAL. Thirty-six brief messages were developed on the basis of the COM-B model and through discussion among the authors, with each message corresponding to one of the four target behaviours and to a different item on the RAL (see Table 1). The messages were typically 100 words long (see S1 Appendix).

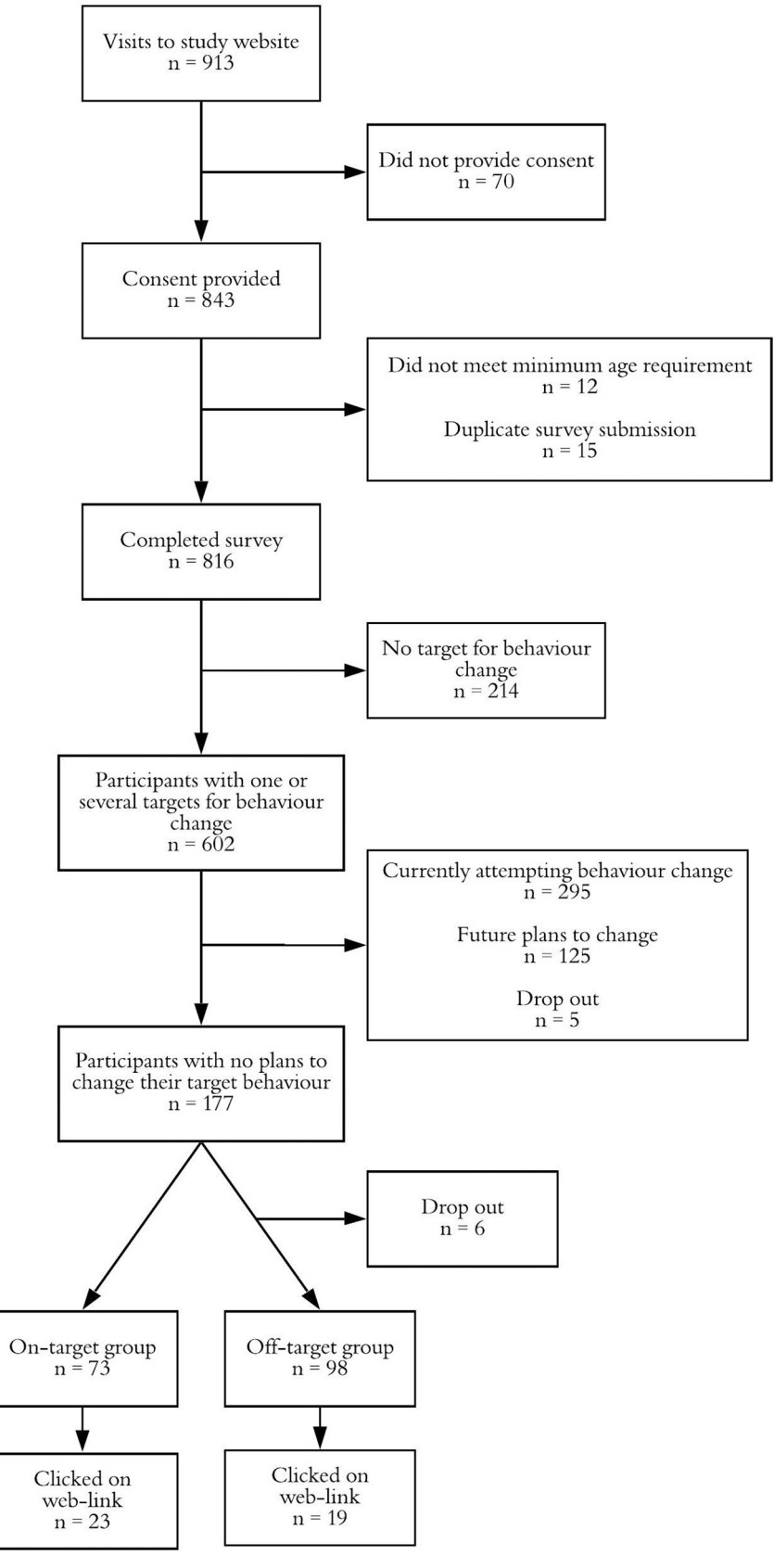

**Fig 1. Procedural flow and participant allocation to the intervention conditions.**

**Table 1. The Risk Acceptance Ladder.**

| I would. . . | COM-B category |
|---|---|
| quit smoking | |
| cut down on the amount that I drink | |
| increase the amount that I am active | |
| increase the amount of fruit and vegetables that I consume | |
| **but**. . . | |
| a) I have not heard that XXXX was harmful or risky | Capability–psychological |
| b) I have heard that XXXX is risky but never fully understood what the problem is | Capability–psychological |
| c) I understand what people are saying about the risks of XXXX, but I don't believe it | Motivation–reflective |
| d) I accept that XXXX is risky but don't care enough to do anything about it | Motivation–automatic |
| e) I think it is worth XXXX, but it is not a priority at the moment | Motivation–reflective |
| f) I don't think I can XXXX because things in my social world make it too difficult | Opportunity–social |
| g) I don't think I can XXXX because things going on in my life make it too difficult | Opportunity–physical |
| h) I don't think I can XXXX because I don't have the self-control | Motivation–automatic |
| i) I want to XXXX, but I don't know how best to do it | Capability–psychological |
| j) Other (none of the above)—Please specify. . . | _ |

**Control.** Participants allocated to the off-target messaging received a randomly selected message from the bank of the 36 brief messages described above that was incongruent with their RAL response.

## Measures

**Covariates.** Data on gender (i.e. male, female, other), age in years and ethnicity (i.e. White, non-White [black, Asian, mixed, other]) were collected at the start of the survey.

**Health behaviours.** Single-item measures were used to assess cigarette smoking status ("*Do you smoke cigarettes at all nowadays*?"), excessive alcohol consumption ("*Do you drink alcohol every day or almost every day*?"), insufficient physical activity ("*Do you make sure that you walk or do other moderate physical activity for at least 30 minutes every day*?") and low fruit and vegetable consumption ("*Do you make sure that you eat at least five portions of fruit and vegetables each day*?"). These were all coded as yes/no. The alcohol item was designed to broadly map onto validated quantity-frequency screening instruments such as the Alcohol Use Disorders Identification Test-Consumption (AUDIT-C) scale, with daily or almost daily drinking classified as excessive alcohol consumption [17, 18]. A single item was used to assess behaviour change plans ("*Which best describes your relationship with [target behaviour]*?"). The response options were: 1) I am seriously trying to (e.g. eat more fruit and vegetables), 2) I have made a definite plan to (e.g. eat more fruit and vegetables) soon and 3) I have no immediate plans to (e.g. eat more fruit and vegetables).

**Risk Acceptance Ladder.** The 10 RAL items related to different aspects of the COM-B model (Capability–four items, Opportunity–two items, Motivation–three items; see Table 1). Participants were encouraged to select one item from the RAL which most closely described what they believed would need to change in order to change their target behaviour, with an additional, non-specific 'Other' option for those who felt none of the nine RAL items were adequate. Those who selected 'Other' were given the opportunity to provide their own reason as a free-text response. The current version of the RAL was arrived at after a number of iterations specifically relating to smoking [7–9].

**Outcomes.** *Performance*. The RAL's performance was determined by two metrics: the percentage of participants who were able to select an applicable RAL response option (i.e. those

who did not select 'Other') and by a question which assessed reported ease of use of the RAL ("*How easy was it to make just one choice*?"). The response options were: 1) very easy, 2) quite easy, 3) not very easy and 4) not at all easy.

*Effectiveness.* A behavioural outcome was used to assess the RAL's effectiveness. Following intervention delivery, participants were provided with a link to a website with information about how to change their target behaviour. The links provided were for the NHS Smokefree website (http://www.nhs.uk/smokefree); the Down Your Drink website (http://www.downyourdrink.org.uk); and the NHS Choices websites for physical activity (http://www.nhs.uk/livewell/fitness/Pages/Fitnesshome.aspx) and healthy eating (http://www.nhs.uk/livewell/5aday/Pages/5ADAYhome.aspx), respectively. A record was made if a participant clicked on the link provided; the act of clicking was interpreted as engagement with the health promotion materials.

### Data analysis

**Qualitative analysis.** Free-text responses to the RAL were analysed by CS and LS with inductive thematic analysis [19], which involved generating initial codes and higher-order themes that captured respondents' underlying beliefs. Responses were coded by CS and double checked by LS, with higher-order themes refined through discussion with RW.

**Quantitative analysis.** Data were analysed in SPSS v.21. Chi-squared and *t*-tests were performed to determine any baseline differences between groups. Descriptive analyses were conducted to summarise the performance indicators (i.e. the ability to self-classify and ease of use of the RAL). A logistic regression analysis, adjusted for gender, age and ethnicity, was used to estimate the association between group allocation and the effectiveness indicator (i.e. clicking on the link provided vs. not clicking on the link).

**Bayes factors.** Given the relatively small sample size and thus low level of power to detect anything other than large effects, classical inferential statistics is limited in the event of non-significant results as it is unable to distinguish between insensitive data and the null hypothesis being correct. For this reason, we planned to analyse the data using a Bayesian approach in the case of non-significant results. The calculation of a Bayes Factor (BF) establishes the relative likelihood of the null versus the experimental hypothesis. Values greater than 3 or smaller than 1/3 are typically regarded as providing substantial evidence for the alternative or null hypothesis, respectively, with intermediate values indicating that data are insensitive to distinguish between the two [20]. BFs were calculated using an online calculator (www.lifesci.sussex.ac.uk/home/Zoltan_Dienes/inference/Bayes.htm) with the alternative hypotheses conservatively represented in each case by a half-normal distribution, where the alternative hypothesis is represented by a population mean of zero and the standard deviation of the distribution specified as an expected, reasonable effect size. This means that plausible values have been effectively represented between zero and twice the effect size, with smaller values represented as being more likely. As no prior data existed on likely effect sizes, we calculated BFs for postulated small (OR = 1.68), medium (OR = 3.47) and large (OR = 6.71) effects [21].

## Results

### Participant characteristics

Of 913 people visiting the study website, 843 (92%) consented to take part in the study. After removing data for people who did not meet the minimum age requirement (*n* = 12) and duplicate survey submissions (*n* = 15), 816 participants were included in the initial analysis (see Fig 1).

The sample was largely White, female and (with an average age of 30 years) relatively young (see Table 2). Respondents had on average 1.2 (SD = 0.9) targets for behaviour change. The most common target was low fruit and vegetable consumption; more than half of respondents reported consuming less than the recommended five portions per day. The least frequent target was cigarette smoking; approximately 1 in 7 respondents reported current smoking (see Table 2).

## Performance

A total of 177 participants had no immediate plans to change their target behaviour. This group was older and more likely to be male than those with plans, but no other differences were observed (see Table 2). Other than the item 'I heard that [insert target behaviour] is risky but never fully understood what the problem was', all RAL items were endorsed at least once. The most commonly endorsed items were 'I think it is worth [changing target behaviour] but it is not a priority at the moment' (28.3%) and 'Other' (31.1%) (see Table 3). Motivation (captured by three RAL items; see S1 Appendix) was the most frequently endorsed COM-B category (65.8%).

**Ability to select appropriate RAL response option.** Over two thirds of participants (68.9%, 95% CI = 61.8%-75.3%) were able to select an appropriate RAL item that identified a key reason why they had not yet changed their target behaviour.

**Ease of use.** Almost two thirds (64.9%, 95% CI = 57.5%%-71.7) of participants found the RAL to be 'very easy' or 'quite easy' to use (see Table 3), but a significantly greater proportion of those who selected 'Other' in response to the RAL stated that it was 'not at all easy' to select one RAL item (14.3% vs. 3.3%; $\chi^2$ (1) = 14.3, $p$ = 0.003). Over half of those who selected 'Other' ($n$ = 29) provided their own reason for not yet changing their target behaviour (see Table 3). The most common themes were that participants disputed that their current behaviour is problematic and that a physical illness or condition prevented behaviour change. These themes align with the COM-B categories of psychological and physical capability, respectively. The remaining themes also addressed issues captured by COM-B categories, such as motivation and opportunity, including enjoyment of the risky behaviour, monetary costs of changing the behaviour, and beliefs about health consequences, such as: "*To give up (smoking), I feel, would put my body into shock, and would probably kill me*".

**Table 2. Participant characteristics.**

| | Full sample ($n$ = 816) | Excluded from pilot RCT ($n$ = 639) | Included in pilot RCT ($n$ = 177) | $p$-value |
|---|---|---|---|---|
| **Gender, % ($n$)** | | | | <0.001 |
| Male | 22.8 (186) | 18.6 (118) | 37.9 (67) | |
| Female | 76.7 (626) | 80.8 (512) | 61.2 (110) | |
| Other | 0.5 (4) | 0.6 (4) | - | |
| **Age, mean ($SD$)** | 30.7 (12.8) | 30.01 (12.31) | 32.9 (14.2) | 0.020 |
| **Ethnicity, % ($n$)** | | | | 0.228 |
| White | 84.4 (689) | 83.8 (531) | 86.4 (153) | |
| Non-White | 15.6 (127) | 16.2 (103) | 13.6 (24) | |
| **Health behaviours[a], % ($n$)** | | | | |
| Cigarette smoking | 14.5(118) | 11.0 (70) | 26.0 (46) | <0.001 |
| Excessive drinking | 13.6 (111) | 10.1 (64) | 26.0 (46) | <0.001 |
| Physical inactivity | 39.7 (324) | 37.9 (240) | 45.8 (81) | 0.035 |
| Low fruit and vegetable consumption | 52.1 (425) | 46.5 (295) | 71.2 (126) | <0.001 |

[a] More than one behaviour could be selected.

**Table 3. Distribution of RAL responses, ease of use and reasons for selecting the 'Other' response option.**

| | % (*n*) |
|---|---|
| **RAL response (*n* = 177)** | |
| A–Unaware of the risks | 6.2 (11) |
| B–Don't fully understand the problem | 0 (0) |
| C–Don't believe the risks | 3.3 (6) |
| D–Don't care enough to change | 9.6 (17) |
| E–Not a priority | 28.3 (50) |
| F–Social environment | 2.8 (5) |
| G–Physical environment | 8.5 (15) |
| H–Self control | 7.3 (13) |
| I–Unsure how | 2.8 (5) |
| J–Other | 31.1 (55) |
| **Ease of use (*n* = 171)** | |
| Very easy | 25.2 (43) |
| Quite easy | 39.8 (68) |
| Not very easy | 28.7 (42) |
| Not at all easy | 6.4 (11) |
| **Other reasons for not changing behaviour (*n* = 29)*** | |
| Disputes that current behaviour is problematic or unhealthy | 48.3 (14) |
| Physical illness or condition (dietary constraints) preventing behaviour change | 24.1 (7) |
| Enjoyment of activity preventing behaviour change | 6.9 (2) |
| Monetary costs preventing behaviour change | 3.4 (1) |
| Social aspects of behaviour preventing change | 3.4 (1) |
| Disbelief of advice preventing behaviour change | 3.4 (1) |
| Belief that they are doing as much as possible | 3.4 (1) |
| Does not get around to following guidelines | 3.4 (1) |
| Belief that their body wouldn't cope with change | 3.4 (1) |

*Those who selected 'Other' were asked to provide further reasons in a free-text box.

## Effectiveness

Following completion of the RAL, participants were randomised to the on- or off-target intervention conditions (*n* = 171). Those who selected 'Other' received off-target messages (as no on-target messages were available), which resulted in a higher proportion of participants allocated to the off-target (control) condition. There were no significant differences in baseline characteristics between those randomised to the on- or off-target conditions (see Table 4).

Nearly twice as many participants allocated to the on-target group (31.5%, *n* = 23) clicked on the link for further information about health behaviour change compared with those allocated to the off-target group (19.4%, *n* = 19), suggestive of an effect in the expected direction (OR = 1.91, 95% CI = 0.95–3.86; *p* = 0.071). The calculation of Bayes factors indicated that our results provided moderate support for the hypothesis of there being a small (BF = 3.2) but not a medium (BF = 2.4) or large (BF = 1.7) effect of the on-target messages. After adjusting for gender, age and ethnicity, those in the on-target group were significantly more likely to click on the link to find out more about how to change their risky health behaviour compared with those in the off-target group (OR = 2.1, 95% CI = 1.0–4.3; *p* = 0.048).

In an unplanned sensitivity analysis, the exclusion of those who selected 'Other' (who were automatically assigned to the off-target group) did not change the direction of the effect, but

**Table 4. Participant characteristics by group allocation (n = 171).**

|  | On-target (*n* = 73) | Off-target (*n* = 98) | *p*-value |
|---|---|---|---|
| **Gender, % (*n*)** |  |  | 0.477 |
| Male | 38.35 (28) | 36.76 (36) |  |
| Female | 61.64 (45) | 63.27 (62) |  |
| Other | - | - |  |
| **Age, mean (*SD*)** | 31.56 (13.20) | 34.17 (15.13) | 0.240 |
| **Ethnicity, % (*n*)** |  |  | 0.452 |
| White | 84.93 (62) | 86.73 (85) |  |
| Non-White | 15.07 (11) | 13.27 (13) |  |
| **Behavioural target, % (*n*)** |  |  |  |
| Cigarette smoking | 17.81 (13) | 18.37 (18) | 0.324 |
| Excessive drinking | 31.14 (22) | 14.29 (14) | 0.124 |
| Physical inactivity | 9.59 (7) | 13.27 (13) | 0.289 |
| Low fruit and vegetable consumption | 42.47 (31) | 54.08 (53) | 0.232 |

the difference did not reach statistical significance (OR = 2.4, 95% CI = 1.0–6.1; *p* = 0.063). Results were substantially unchanged after adjustment for gender, age and ethnicity (OR = 2.2, 95% CI = 0.9–5.9; *p* = 0.1). The calculation of Bayes factors indicated that our results provided moderate support for the hypothesis of there being a small (BF = 3.3) and medium (BF = 3.1) but not a large (BF = 2.4) effect.

## Discussion

This study examined the potential effectiveness and ease of use of an online version of the RAL in promoting help-seeking for a range of health behaviours. The RAL appeared to be relatively easy to complete by the participants, with most participants being able to select a single 'rung' on the ladder. Tailoring messaging to the selected rung may have increased the likelihood that the participant would take an initial step towards changing their behaviour. This finding adds to the currently mixed evidence on the effectiveness of health interventions tailored to participants' motivational stage, assessed at baseline [12–15].

As over two thirds of participants were able to successfully use the RAL to classify the source of their inability to change, and a similar proportion found the RAL easy to use, it would seem that this new measure has good usability. The scalability of the RAL and the targeted messaging is also promising; once targeted messages have been developed, they can be delivered to a large number of people at the click of a button. This requires few resources and minimal input from trained staff. Hence, the RAL might be useful for clinicians and policy makers who wish to assess reasons for health behaviour inertia and to prompt engagement with health behaviour change interventions.

When asking participants who selected 'Other' on the RAL to provide their own reason for not changing, the most commonly provided response was that participants did not think that their current behaviour, or the level at which they were currently performing a particular behaviour, was unhealthy or problematic. This theme closely relates to existing items on the RAL (i.e. the first two items), and maps onto the construct of psychological capability in the COM-B model. Nearly half of the participants who selected 'Other' fell into this category. Rewording the existing two items to more closely reflect people's understanding of its content (e.g. by means of cognitive interviews) may therefore help people self-classify more easily on the RAL. However, some free-text responses were not easily captured by existing RAL items, most notably the inability to change behaviour due to a physical illness or condition (endorsed

by a quarter of participants who selected 'Other'). This previously unaddressed theme relates to physical capability in the COM-B model and an additional item to capture this issue (e.g. "I don't think I can XXXX because I am not physically able to do so" or "I don't think I can XXXX because it will make me physically uncomfortable in some way") has the potential to add to the value of the RAL. Thus, further qualitative work is required to explore a wider range of RAL items. Moreover, further research using the RAL would benefit from allowing participants to rate the messages they received in terms of perceived personal relevance and interest. Following further development of the measure, our results indicate that the evaluation of the RAL-based targeted messaging in a fully powered RCT may be warranted.

## Limitations

This study had several limitations. First, as this pilot study was not pre-registered, the results should be considered exploratory. Second, there was a lack of early user involvement in the development of the RAL. As mentioned above, further qualitative work is therefore needed to explore a wider range of RAL items. Third, due to the small sample size, it was not possible to estimate whether the targeted messages were equally effective for all health behaviours. Fourth, although the on-target intervention was found to increase subsequent engagement with health messaging, it is unclear whether this engagement translates to actual behaviour change. Fifth, respondents who selected 'Other' on the RAL received off-target messages, with sensitivity analyses conducted to assess whether the direction of the effect remained robust when excluding this group. Although the exclusion of those who selected 'Other' did not alter the direction of the effect, the difference no longer reached statistical significance. This may be due to low power for this comparison or be indicative of a potential bias whereby those selecting 'Other' were consistently less motivated to change compared with those who were able to select a single 'rung' on the RAL. Sixth, our recruitment primarily targeted university students, with the resulting sample being predominantly White, female and relatively young (i.e. an average age of ~30 years). This likely limits the generalisability of the results to the general population without current plans to change the selected health behaviours. Finally, as the study was conducted in 2015, future applications of the RAL may benefit from updating the website design to ensure it aligns with users' evolving expectations.

## Conclusion

The RAL could be a useful tool for targeting messaging around increasing self-protective behaviours. Further research is required to improve the RAL and extend its evaluation to clinically meaningful outcomes and additional types of behaviour.

## Supporting information

**S1 Appendix. Description of the recruitment materials, screening questionnaire and the Risk Acceptance Ladder.**
(DOCX)

## Author Contributions

**Conceptualization:** Claire Stevens, Robert West, Lion Shahab.

**Formal analysis:** Claire Stevens, Lion Shahab.

**Investigation:** Claire Stevens, Lion Shahab.

**Methodology:** Claire Stevens, Robert West, Lion Shahab.

**Writing – original draft:** Olga Perski, Lion Shahab.

**Writing – review & editing:** Olga Perski, Lion Shahab.

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
