## [Decision Letter · Decision Letter 0]

9 Mar 2021

PONE-D-20-33095

Pilot randomised controlled trial of the Risk Acceptance Ladder (RAL) as a tool for targeting health communications

PLOS ONE

Dear Dr. Lion,

Thank you for submitting your manuscript to PLOS ONE. After careful consideration, we feel that it has merit but does not fully meet PLOS ONE’s publication criteria as it currently stands. Therefore, we invite you to submit a revised version of the manuscript that addresses the points raised during the review process.

Kindly adhere to the comments given by myself and the reviewers.

We look forward to receiving your revised manuscript.

Best,

MI Subhani, PhD.

Academic Editor

PLOS ONE

Journal Requirements:

"I have read the journal's policy and the authors of this manuscript have the following competing interests: OP and CS report no conflicts of interest. RW has undertaken research and consultancy for and receives travel funds and hospitality from manufacturers of smoking cessation medications (Pfizer, GlaxoSmithKline and Johnson and Johnson). LS has received a research grant and honoraria for a talk and travel expenses from manufacturers of smoking cessation medications (Pfizer and Johnson & Johnson)."

Additional Editor Comments:

Write the Abstract according to the following algorithm: first two or three sentences indicate the relevance of the topic; the aim and object of the study; the methodology (methods) of the study (for theoretical studies – its theoretical basis) are described; the obtained results and their practical value are characterized. Dedicate most of the Abstract to the result. The volume of the Abstract is 200-250 words.

Reviewers' comments:

Reviewer's Responses to Questions

**Comments to the Author**

1. Is the manuscript technically sound, and do the data support the conclusions?

Reviewer #1: Yes

Reviewer #2: Yes

2. Has the statistical analysis been performed appropriately and rigorously? 

Reviewer #1: Yes

Reviewer #2: Yes

3. Have the authors made all data underlying the findings in their manuscript fully available?

Reviewer #1: Yes

Reviewer #2: Yes

4. Is the manuscript presented in an intelligible fashion and written in standard English?

Reviewer #1: Yes

Reviewer #2: Yes

5. Review Comments to the Author

Reviewer #1: Review PLOS ONE - Risk Acceptance Ladder

This article describes a study exploring the use and potential effectiveness of an electronic questionnaire and automated response to promote help seeking for four risky health behaviors (smoking, alcohol consumption, physical activity, and fruit/vegetable consumption). Among 816 participants included, 602 had at least on risky behavior, and 28% of these had no plan to change this/these behavior(s) and took part in the study. About 2/3 of participants were able to select an applicable item on the tested scale (the Risk Acceptance Ladder - RAL). Also, about 2/3 of participants reported that it was easy to select one option. Participants were then randomized to receive a short message, which was either congruent or incongruent with their RAL response. Compared with the incongruent group, participants allocated to the congruent group had greater odds of clicking on a link to receive further information about this risky behavior.

Overall, this study is scientifically sound and was carefully conducted. The reporting is of excellent quality in general. Nevertheless, a few caveats are to be considered to strengthen this manuscript. Those are listed below.

p.5 lines 118-129, Procedure. This section describes the participants flow. Part of it is redundant with the first section of the results (p. 9-10).

p.8, line 211, “Free-text responses were coded and summarised in line with standard thematic analysis (17)”. Authors should develop their description of this analysis. Referring to a methods article is not sufficient. Qualitative analysis should be indicated as a sub-heading and details on how it was conducted should be provided. How exactly were data coded? Who did code the responses? Was there double coding? Any triangulation of the data and related coding? Also, authors should discuss the potential limitations of their analytic perspective in the relative discussion section.

The discussion is quite succinct and does not put the findings in perspective with the literature in the field.

Discussion, 2nd paragraph. Authors should discuss the small effect size and the low proportion of participants “engaging” further (31%, respectively 19%, accessed a website, but we do not know how far they went on this website).

Reviewer #2: This paper is clear and well written. It reports on a potentially useful approach for targeting health behaviour change with patients in time constrained settings.

A few suggestions to enhance the completeness of the paper:

- Recruitment targeted university students in one avenue, did this limit the demographics of the sample – they were predominantly white, were they also educated, and what are the potential implications of this?

- The data is quite old, having been collected in 2015. I’m not sure if that is relevant but it seems a long gestation time for this study.

- The measure of ‘excessive alcohol consumption’ seems somewhat crude – drinking every day is not in itself excessive if the daily consumption is limited to less than two standard drinks – perhaps the authors could provide some explanation/justification for the health behaviour measures chosen.

- The high rate of endorsement of ‘other’ seems to suggest that the RAL is missing important patient beliefs which could be incorporated in an adjusted version. The authors mention possible re-wording but could provide more specific examples of potential future adaptations.

- The authors have conducted appropriate sensitivity analysis to capture the potential bias, nut could be more explicit in the discussion in addressing the fact that respondents selecting ‘other’, who all cited negative attitudes towards behaviour change or disbelief of the risks of the behaviour, were allocated to off target messages – this could bias the results in that the motivation levels of these individuals would appear to be lower than that of people who can least appreciate the value of the indicated behaviour change.

- The limited nature of outcome assessment should be specifically addressed in the discussion – the impact of targeted messaging is limited to clicks to relevant websites which is an immediate measure only, and does not indicate intentions or actual behaviour change.

6. PLOS authors have the option to publish the peer review history of their article (what does this mean?). If published, this will include your full peer review and any attached files.

Reviewer #1: No

Reviewer #2: No

---

## [Author Response · Author response to Decision Letter 0]

29 Mar 2021

See attached word document that details our response to reviewers.

---

## [Decision Letter · Decision Letter 1]

18 Oct 2021

PONE-D-20-33095R1Pilot randomised controlled trial of the Risk Acceptance Ladder (RAL) as a tool for targeting health communicationsPLOS ONE

Dear Dr. Shahab,

Thank you for submitting your manuscript to PLOS ONE. After careful consideration, we feel that it has merit but does not fully meet PLOS ONE’s publication criteria as it currently stands. Therefore, we invite you to submit a revised version of the manuscript that addresses the points raised during the review process.

 Our guidelines for data availability (https://journals.plos.org/plosone/s/data-availability) indicate that: For studies analyzing data collected as part of qualitative research, authors should make excerpts of the transcripts relevant to the study available in an appropriate data repository, within the paper, or upon request if they cannot be shared publicly. If even sharing excerpts would violate the agreement to which the participants consented, authors should explain this restriction and what data they are able to share in their Data Availability Statement.As your manuscript only contains one quote, we don't think that this has been met. We request that you update your data availability statement and indicate whether the qualitative data can be shared, and where they can be obtained. Please submit your revised manuscript by Nov 29 2021 11:59PM. If you will need more time than this to complete your revisions, please reply to this message or contact the journal office at plosone@plos.org. Please include the following items when submitting your revised manuscript:A rebuttal letter that responds to each point raised by the academic editor and reviewer(s). You should upload this letter as a separate file labeled 'Response to Reviewers'.A marked-up copy of your manuscript that highlights changes made to the original version. You should upload this as a separate file labeled 'Revised Manuscript with Track Changes'.An unmarked version of your revised paper without tracked changes. You should upload this as a separate file labeled 'Manuscript'.If applicable, we recommend that you deposit your laboratory protocols in protocols.io to enhance the reproducibility of your results. Protocols.io assigns your protocol its own identifier (DOI) so that it can be cited independently in the future. For instructions see: https://journals.plos.org/plosone/s/submission-guidelines#loc-laboratory-protocols. Additionally, PLOS ONE offers an option for publishing peer-reviewed Lab Protocol articles, which describe protocols hosted on protocols.io. Read more information on sharing protocols at https://plos.org/protocols?utm_medium=editorial-email&utm_source=authorletters&utm_campaign=protocols.

We look forward to receiving your revised manuscript.

Kind regards,

Yann Benetreau, PhD

Senior Editor, *PLOS ONE*

Journal Requirements:

Reviewers' comments:

Reviewer's Responses to Questions

**Comments to the Author**

1. If the authors have adequately addressed your comments raised in a previous round of review and you feel that this manuscript is now acceptable for publication, you may indicate that here to bypass the “Comments to the Author” section, enter your conflict of interest statement in the “Confidential to Editor” section, and submit your "Accept" recommendation.

Reviewer #1: All comments have been addressed

2. Is the manuscript technically sound, and do the data support the conclusions?

Reviewer #1: Yes

3. Has the statistical analysis been performed appropriately and rigorously? 

Reviewer #1: Yes

4. Have the authors made all data underlying the findings in their manuscript fully available?

Reviewer #1: Yes

5. Is the manuscript presented in an intelligible fashion and written in standard English?

Reviewer #1: Yes

6. Review Comments to the Author

Reviewer #1: Authors adequately addresed reviewers' comments and the current manuscript has been improved and would be ready for publication.

7. PLOS authors have the option to publish the peer review history of their article (what does this mean?). If published, this will include your full peer review and any attached files.

Reviewer #1: No

---

## [Author Response · Author response to Decision Letter 1]

21 Oct 2021

N/A - this revision only included a request for a technical change in Data Availability Statement. Reviewers were satisfied with previous responses and did not request further changes.

---

## [Editor Report · Decision Letter 2]

2 Nov 2021

Pilot randomised controlled trial of the Risk Acceptance Ladder (RAL) as a tool for targeting health communications

PONE-D-20-33095R2

Dear Dr. Shahab,

We’re pleased to inform you that your manuscript has been judged scientifically suitable for publication and will be formally accepted for publication once it meets all outstanding technical requirements.

Sincerly yours,

Yann Benetreau, Ph.D.

Senior Editor, *PLOS ONE*
---

## [Editor Report · Acceptance letter]

4 Nov 2021

PONE-D-20-33095R2 

Pilot randomised controlled trial of the Risk Acceptance Ladder (RAL) as a tool for targeting health communications 

Dear Dr. Shahab:

I'm pleased to inform you that your manuscript has been deemed suitable for publication in PLOS ONE. Congratulations! Your manuscript is now with our production department. 

Kind regards, 

on behalf of

Dr. Yann Benetreau 

Staff Editor

PLOS ONE